# Back in the Driver's Seat: How New EU Greenhouse-Gas Reporting Schemes Challenge Corporate Accounting

Julian Baehr [1,*], Florian Zenglein [2], Guido Sonnemann [3], Markus Lederer [2] and Liselotte Schebek [1]

1   Institute IWAR Material Flow Management and Resource Economy, Technische Universität Darmstadt, 64287 Darmstadt, Germany; l.schebek@iwar.tu-darmstadt.de
2   Department of International Relations, Institute for Political Science, Technische Universität Darmstadt, 64283 Darmstadt, Germany; florian.zenglein@tu-darmstadt.de (F.Z.); lederer@pg.tu-darmstadt.de (M.L.)
3   Institute of Molecular Sciences, University of Bordeaux, Centre National de la Recherche Scientifique, Bordeaux INP, ISM, UMR 5255, 33400 Talence, France; guido.sonnemann@u-bordeaux.fr
*   Correspondence: j.baehr@iwar.tu-darmstadt.de

**Abstract:** Greenhouse-gas (GHG) reporting schemes for companies are increasingly part of climate-mitigation policies worldwide. Notably, the European Green Deal (2019) boosts new public regulations that oblige companies to compile GHG emission inventories, i.e., account for their emissions in a given system boundary. Along with this boost, the workload for companies increases; at the same time, the quality of reporting is questioned. Given the overarching goal to improve companies' climate-mitigation performance, the quality of reporting is inseparably connected to the quality of the respective accounting. However, the literature discusses carbon accounting as a universal umbrella term focusing on managerial issues, thus disregarding the crucial role of accounting methodologies in the sense of calculation approaches. In this publication, we apply an analytical approach introducing a clear differentiation between the task of quantitatively accounting for GHG inventories and the task of reporting results from calculated inventories in response to stakeholder or policy expectations. We use this approach to investigate European GHG reporting schemes and related GHG accounting methodologies in detail. Our findings indicate that the current phase of the European Green Deal depicts a quantitative growth in reporting schemes and a significant qualitative change by shifting from formerly voluntary to mandatory reporting schemes, along with the application of accounting methodologies originally not intended for politically compulsory purposes. We analyze the consequences of this shift, which poses new challenges for companies and policymakers, i.e., data-management concepts and refined methodological frameworks.

**Keywords:** carbon accounting; European Green Deal; industrial ecology; net-zero; reporting scheme; sustainable policy

## 1. Introduction

Today, companies, as major emitters of greenhouse gases (GHG), must disclose the GHG emissions of their products, production installations, or entire organizations to comply with existing regulations and provide transparency to investors, shareholders, and other stakeholders. This disclosure of information, notwithstanding a possible mandatory or voluntary character, is summarized under the term (*corporate*) *GHG reporting*. The basic requirement for such reporting is the compilation, or accounting, of a GHG emission inventory, yielding quantitative information on GHG emissions or GHG performance-related indicators. While methodologies ensuring consistent accounting of disclosures first evolved on a national level [1] the development of methodologies on a corporate and product level followed and is picking up unprecedented speed. In Europe, this speed has been significantly induced by the European Green Deal [2], which has triggered the development of multiple new public reporting regulations, such as the EU Taxonomy [3] or the upcoming Ecodesign for Sustainable Products Regulation (ESPR) [4]. The European Commission

systematically imposes increasing regulatory GHG reporting pressure on companies, thus putting itself back in the driver's seat and exhibiting "*finally real leadership*" in the climate-policy arena [5]. Accordingly, a boost of methodological enhancement and related technical documents for GHG inventory accounting can be seen, partly already published and partly to be expected in the near future. As a result of the current political momentum, companies struggle to keep up with the latest reporting requirements and experience increasing accounting difficulties regarding the choice of methodology, its procedural peculiarities, and the practical acquisition of available, appropriate, and valid data. The resulting plethora of reporting schemes significantly increases the practitioners' confusion, leading to increased workload and methodological and practical accounting errors that negatively affect the reporting quality. The recent literature shows that the quality of carbon information is not necessarily connected to the increase in GHG reporting [6]. On the contrary, systematic errors like reporting inconsistency, boundary incompleteness, and activity exclusion may omit as much as 50% of the disclosed GHG emissions [7]. The authors argue that these systematic errors are a result of "*current carbon accounting and reporting practices* [which] *remain unsystematic and not comparable, particularly for emissions along the value chain*" (p. 6).

Although the term corporate GHG reporting is omnipresent in climate issues, no systematic investigation of its definition or interpretation in different contexts is found in the academic literature. Thus, the congruence in the literature lies in the general meaning of reporting, whether financial or non-financial, which denotes the activity to provide or disclose information about a particular reporting entity to stakeholders [8]. In contrast, three significant literature reviews have explored carbon accounting by analyzing 129, 117, and 137 studies published between 1994 and 2022 [9–11]. They consistently reveal that carbon accounting is a heterogenic term that, from the stakeholder perspective, includes diverse and multi-purpose responsibilities. Stechemesser and Guenther [9] state that carbon accounting can be defined as "*the measuring, collation, assessment and communication (...) and the monetary valuation of GHG emissions (as assets and liabilities) to provide this information to internal or external audiences*" (p. 25). Similarly, He et al. [10] and Hazaea et al. [11] build on a definition proposed by Tang [12] and conclude: "*there is a consensus that corporate carbon accounting refers to the use of accounting methods to collect, analyse, verify and report climate change information, account for carbon assets and liabilities, manage carbon risks and evaluate carbon performance for more informed decision-making by managers and external users*" [10] (p. 288). Thus, carbon accounting is used in the literature as a universal umbrella term and is understood as a management concept on how to deal with "*unbooked liabilities*" of future compliance and carbon mitigation costs [10] (p. 284). The existing literature is mainly driven by managerial issues regarding compliance costs, risks, and financial performance [13,14]. This management perspective is also mirrored by the large extent of governance incentives, strategy, and risk information required by regulations such as the Corporate Sustainability Reporting Directive (CSRD) [15]. However, reporting necessarily requires valid information on systematically quantified emissions to improve a company's performance in line with climate-policy objectives. Therefore, the quality of reporting is inseparably connected to the quality of the respective GHG inventories and, thus, its underlying accounting methodology, as well as the availability and quality of the utilized data sources. Tang [12] acknowledged this relationship and distinguished the broader carbon accounting term with its managerial implications from the technical procedures to physically calculate GHG emissions, which he called "*GHG accounting*" (p. 10).

Despite the strong link between GHG reporting and GHG accounting, only a few studies have investigated their interactions and related data acquisition. Bowen and Wittneben [16] introduced three organizational fields: first, counting carbon on a molecular level, denoting data acquisition; second, carbon accounting on an organizational level, denoting the calculation process; and third, accountability for carbon on a global level, denoting disclosure. In relation to the second field, sustainability-management accounting was proposed to increase the efficiency of GHG-related information collection [17]. This concept was later applied in practice by determining what, how often, where, and why the

information was collected [18], and case studies were performed to account for corporate and product emissions in a combined approach [19,20]. While these studies primarily aimed at reducing the accounting workload, several studies identified the voluntary nature of existing GHG reporting schemes as a key reason for low-quality disclosures. They, thus, suggested mandatory reporting regulations to align methodological and data-related peculiarities [21–23].

To develop strategies for improving the quality of GHG disclosures, we hypotigize that the problem of unsystematic and non-comparable practices has to be tackled as a first step. Therefore, understanding the procedures and methodological foundations of GHG reporting and GHG accounting is indispensable. Based on this differentiation, the interrelation of these terms can be investigated, leading to a holistic but analytically substantiated problem understanding. In this publication, we, therefore, apply an analytical approach (specified in Section 2) that decomposes the field of carbon accounting and reporting practices by separately describing their historical evolution, classifying key representatives, and analyzing their specific characteristics (Section 3). These characteristics substantiate a detailed discussion of specific interactions between reporting and accounting in different dimensions while maintaining the holistic perspective governing reporting quality (Section 4). Finally, Section 5 summarizes the analytical results and draws conclusions on possible future applications to which these results can contribute. Accordingly, our study aims to identify and distinguish major accounting and reporting concepts and, thus, structure the landscape of the ever-growing plethora of denominations and specifications of approaches. Thus, we intend to help practitioners better understand the desired outcome of reporting requirements, as well as its methodological implications, contributing to He et al.'s [10] call for *"greater theoretical sophistication"* to *"broaden and deepen [the] understanding of carbon accounting issues"* (p. 286).

## 2. Materials and Methods

Our analytical approach followed a systematic working procedure that included the definition of terms; the separate classification and characterization of reporting and accounting; and the multidimensional discussion of interactions.

Definition of Terms.

To decompose the carbon accounting term, we introduce the following definitions:

First, following Tang [12], we apply the term *GHG accounting* to denote carbon accounting in a narrower sense as obtaining input data from various data sources to calculate a GHG emission inventory within defined organizational, geographical, temporal, and technical system boundaries. Accordingly, a *GHG accounting methodology* describes technical parameters like system boundaries, methodological procedures, and calculation rules. Second, we define *GHG reporting* as disclosing output data of priorly accounted inventories, either on its own or embedded in additional information, typically in a standardized format, and always aimed at an outside audience, such as regulatory bodies, investors, or other stakeholders. Accordingly, a *reporting scheme* defines the reporting format, respective reporting requirements, and accounting methodology. Third, for analyzing the development of reporting, we introduce the term *public regulation* to acknowledge the strong link between regulations and reporting requirements: Public regulations link an identified problem with the desired outcome by encompassing specific measures, tools, or methods legally set by public authorities on the national or international level and aimed at achieving specific policy objectives. These can comprise an array of measures, including legislative and regulatory actions and technological standards. In our context, public regulations delineate elements such as the reporting objective, the desired outcome, and the recipients involved and establish whether they carry a mandatory or voluntary nature. Thus, they create various forms of direct or indirect pressure on companies, compelling them to accurately account for and report their actions.

Classification and Characterization of Reporting and Accounting.

Based on these definitions, we separately analyzed reporting schemes and accounting methodologies following a three-step procedure: (i) we reviewed the historical development and classified relevant representatives; (ii) based on the classification, we developed criteria to identify key characteristics; and (iii) based on the first two steps, we derived and summarized analytical findings to substantiate the discussion in Section 4. The analysis evaluated policy documents, public regulations, normative standards or guidelines, and the scientific literature. Therefore, the corpus of analyzed documents encompassed the respective legal documents, their technical annexes, amendments, implementing regulations, delegated acts, other related mandatory or voluntary normative standards, and information provided on the websites of the European Commission. The scientific literature was obtained from "The Web of Science", "Scopus" and "Google Scholar", using and combining the research terms "carbon", "GHG", "accounting", "reporting", "regulation", "policy,", "Green Deal", "data sources", "data needs" and "data management". Due to the European Green Deal's overall objective of achieving no net GHG emissions in 2050 [2], and the broad legislative framework it encompasses, we constrained our analysis by focusing entirely on GHG. Accordingly, we excluded other environmental, social, and governance indicators and reporting-related qualitative information, like metadata. Furthermore, we disregarded all reporting schemes, which might have been introduced to reduce GHG emissions but do not directly require GHG accounting, e.g., energy-related reporting. Since we focused primarily on current developments made within the European Green Deal, we disregarded non-European legislation and other information that exclusively applies to companies outside the EU.

Multidimensional Discussion of Interactions.

Based on this analysis, we discussed the interrelation of GHG reporting and accounting, including implementing data sources for data-management systems in several dimensions. This discussion was based on a detailed analysis using typologies from political science and the methodological accounting literature; methodological and structural criteria like reporting entities, target audience, desired reporting outcome, and accounting parameters, such as assessed system boundaries considered GHG; and data sources consulted in practice. Using these characteristics, we finally arrived at a detailed description of interrelations for different dimensions and a holistic perspective.

## 3. Results

### 3.1. Analysis of Reporting Schemes

3.1.1. Classification and Review of Historical Development

According to Luo et al. [24] and He et al. [10], the relevance of GHG reporting for companies is governed by both internal factors (corporate governance and financial resources) and external pressures (social, economic, regulatory, and financial market-driven pressures). However, while they do not prioritize the relevance of these pressures and present them as equally influential, other studies emphasize the relevance of regulatory pressure by highlighting the general role of regulatory bodies [25] and other policies [25].

Since the relevance of regulatory pressure becomes particularly apparent within the new legislative framework of the European Green Deal, we provide a short overview of how GHG reporting evolved, focusing mainly but not entirely on regulatory developments. We differentiate between (i) *intergovernmental* schemes that indirectly lead to further reporting requirements; (ii) *self-regulatory* schemes that are part of a broader trend towards environment, social, and governance (ESG) reporting; (iii) *more mandatory attempts* with comprehensive requirements for the disclosure of environmental matters and enhanced transparency in social and environmental reporting; and (iv) *new regulatory momentum* that reinforces a shift to attain climate neutrality by 2050 in line with the European Green Deal (Figure 1). These four phases should not be seen as describing a historical evolution; rather, they have co-developed into a form of layering where new reporting schemes come on top of each other.

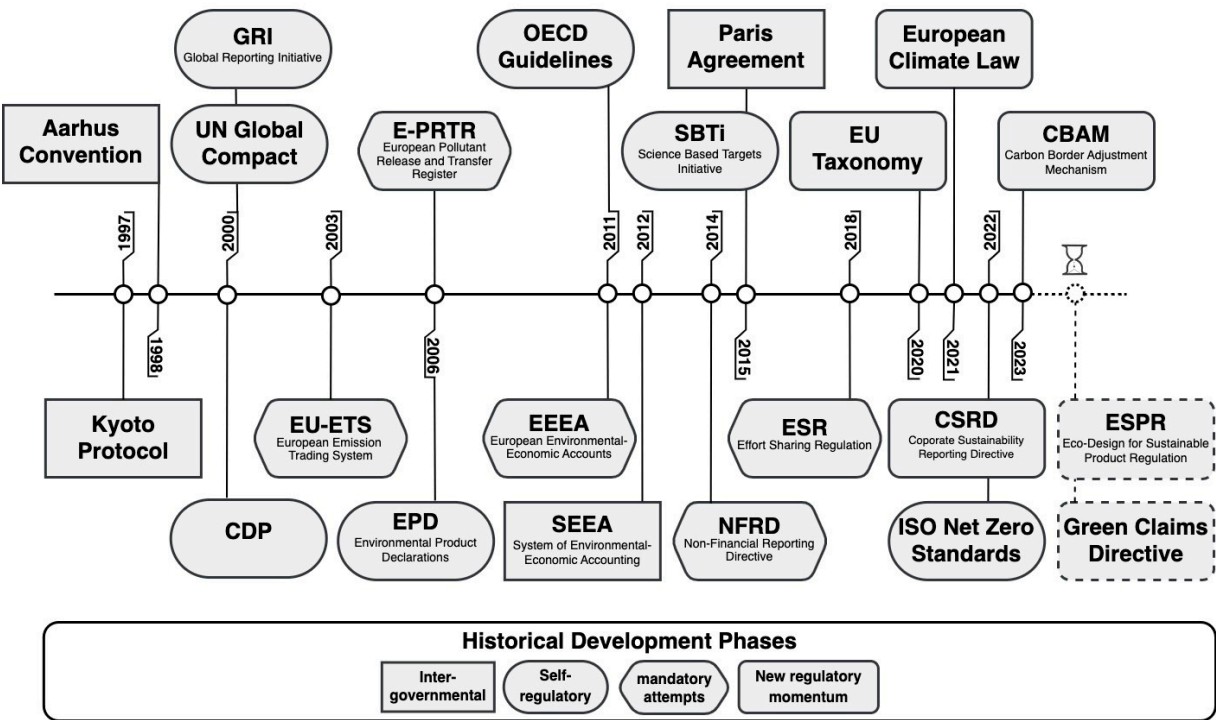

**Figure 1.** Evolution of GHG reporting schemes applying to European stakeholders (if a reporting scheme is defined within a public regulation, the regulation is indicated).

On the intergovernmental level, GHG reporting refers to the Kyoto Protocol [26], when governments in industrialized states started to report their national GHG inventories. While the Kyoto Protocol only committed industrialized countries, the Paris Agreement [27] commits all countries to the Convention to limit global warming to below 2°, and preferably to 1.5°, Celsius compared to industrial levels. The commitments to reducing global GHG emissions gave rise to additional intergovernmental conventions and frameworks, such as the Aarhus Convention [28] and the "System of Environmental-Economic Accounting" (SEEA), which has revised its Central Framework (SEEA-CF) since 1993, adopting its latest version in 2012 and publishing it in 2014 [29,30].

These developments brought a significant shift in the corporate landscape, prompting organizations to establish self-regulatory reporting schemes [31,32]. Organizational self-regulatory GHG reporting mainly developed within broader ESG reporting schemes and resulted in multiple guidelines and standards, such as the Global Reporting Initiative (GRI) standards, the UN Global Compact, the CDP, or the Organization for Economic Co-operation and Development Guidelines for Multinational Enterprises (OECD Guidelines) [33–36]. Further self-regulatory schemes have been developed to define and keep track of GHG mitigation pathways, e.g., ISO Net Zero standards and Science-based Targets initiative [37,38]), and to report product emissions, such as Environmental Product Declarations (EPDs) [39].

Over time, the EU introduced more mandatory attempts to implement commitments made within intergovernmental agreements. These included the implementation of the European Emissions Trading System (EU-ETS) [40], as demanded by the Kyoto Protocol; the European Pollutant Release and Transfer Register (E-PRTR) [41], to reflect the objectives of the Aarhus Convention; the European Environmental–Economic Accounts (EEEA) [42], to derive environmental statistics in line with the SEEA framework; and the Effort Sharing Regulation (ESR) [43], to comply with the reduction goals of the Paris Agreement. By introducing the Non-Financial Reporting Directive (NFRD) [44], the EU built upon prior self-regulatory developments mandating ESG reporting for selected large companies by referring to established and recognized reporting standards.

With the introduction of the European Green Deal and the overall goal to attain climate neutrality by 2050, as outlined in the European Climate Law [45], GHG reporting gained an unprecedented new regulatory momentum. Consequently, multiple regulations have been newly introduced and amended: the EU Taxonomy introduced GHG reporting obligations on selected products and activities to redirect investments, CSRD amended NFRD by obliging more companies to report and introduced mandatory European Sustainability Reporting Standards, and the Carbon Border Adjustment Mechanism (CBAM) [46] introduced GHG reporting obligations for non-EU companies to prevent carbon leakage. In addition, further legislation is expected to be introduced soon. The ESPR [4] will require comprehensive product-related disclosures within digital product passports, and the Green Claims Directive [47] will foreseeably introduce criteria for substantiating environmental claims.

### 3.1.2. Characterization

Due to the dominant regulatory pressure, we reject all self-regulatory schemes and restrict our further analysis to public regulations and their corresponding reporting schemes. Table 1 indicates descriptive characteristics for both and attributes them to the development phases defined above. Methodological characteristics (who, how often, what, to whom, and where) exclusively apply to the reported content in which accounting results are disclosed, i.e., quantitative GHG emission information. In this regard, we also state if the recipient of reporting provides reported output data within an openly accessible database. Furthermore, we characterize the desired reporting outcomes, which indicate the rationale or underlying motivation of a reporting obligation (why). Therefore, we introduce the following categories to display generic motivation types (and justify our categorization by referencing the respective recital or article):

- Monitoring for compliance applies to public regulations that monitor compliance to reduction obligations;
- Information provision for evidence-based policymaking applies to public regulations that provide statistical GHG data for policymaking without monitoring compliance to reduction obligations;
- Facilitation of public participation applies to public regulations that provide GHG data to inform the general public without monitoring compliance to reduction targets;
- Reorientation of capital flows by economic means applies to public regulations that aim to redirect monetary flows into sustainable activities;
- In addition, we use the category individual for public regulations with motivations that are more specific (i.e., "individual" for a certain regulation) rather than the previously defined generic categories.

**Table 1.** Characterization of Reporting Schemes.

| Public Regulation—Descriptive Characterization | | | | Reporting Scheme—Descriptive Characterization | | | Historical Development | Methodological Characterization | | | | | Characterization of Desired Outcome (WHY) | |
|---|---|---|---|---|---|---|---|---|---|---|---|---|---|---|
| Name | Short Name | First Adopted | Legislative Document (Initial Version) | Name | Short Name | Technical Document (Currently Valid Version) | Phase | Reporting Entity (WHO) | Frequency (HOW OFTEN) | Recipient (TO WHOM) | Reported Result (WHAT) | Output Database (WHERE) | Category | Citation (Reference/Recital) |
| Kyoto Protocol | – | 11/12/97 | Kyoto Protocol to the United Nations Framework Convention on Climate Change (U.N. Doc FCCC/CP/1997/7/Add.1) | National Inventory Report | NIR | Annex I Decision 24/CP.19: Revision of the UNFCCC reporting guidelines on annual inventories for Parties included in Annex I to the Convention | Intergovernmental | Member State | Annually | UNFCCC | Annual emissions are divided into sectors, categories, and subcategories | [48] | Monitoring for compliance | *"achieve quantified emission limitation and reduction commitments under Article 3"* (Article 2) |
| Paris Agreement | – | 12/12/15 | Conference of the Parties, Adoption of the Paris Agreement (U.N. Doc. FCCC/CP/2015/L.9/Rev/1) | National Inventory Report | NIR | Annex I Decision 24/CP.19: Revision of the UNFCCC reporting guidelines on annual inventories for Parties included in Annex I to the Convention | Intergovernmental | Member State | Annually | UNFCCC | Annual emissions are divided into sectors, categories, and subcategories | [48] | Monitoring for compliance | *"to hold the increase in the global average temperature to well below 2 °C (...) and pursuing efforts to limit the temperature increase to 1.5 °C above pre-industrial levels"* (Article 2) |
| UNECE Convention on Access to Information, Public Participation in Decision-making, and Access to Justice in Environmental Matters | Aarhus Convention | 25/06/98 | Convention on Access to Information, Public Participation in Decision-Making, and Access to Justice in Environmental Matters | Pollutant Release and Transfer Register | PRTR | Article 7 of the Kyiv Protocol on Pollutant Release and Transfer Registers | Intergovernmental | Companies | Annually per installation | National competent authority | Total annual emissions/installation | [49] | Facilitation of public participation | *"each Party shall guarantee the rights of access to information (...) in environmental matters"* (Article 1) |
| System of Environmental–Economic Accounting | SEEA | 02/03/12 | Official Records of the Economic and Social Council, 2012, Supplement No. 4 (E/2012/24), chap. I.B decision 43/105 | System of Environmental–Economic Accounting 2012 Central Framework | SEEA-CF | System of Environmental–Economic Accounting 2012 Central Framework | Intergovernmental | Eurostat/European Union | Annually | Organization for Economic Co-operation and Development (OECD) | Annual emissions are divided into sectors and categories | [50] | Information provision for evidence-based policymaking | *"to provide integrated information for evidence-based policymaking"* (SEEA-CF Preface by the Secretary-General of the United Nations) |

**Table 1.** *Cont.*

| | Public Regulation—Descriptive Characterization | | | | Reporting Scheme—Descriptive Characterization | | | Historical Development | Methodological Characterization | | | | | Characterization of Desired Outcome (WHY) | |
|---|---|---|---|---|---|---|---|---|---|---|---|---|---|---|---|
| Name | Short Name | First Adopted | Legislative Document (Initial Version) | Name | Short Name | Technical Document (Currently Valid Version) | Phase | Reporting Entity (WHO) | Frequency (HOW OFTEN) | Recipient (TO WHOM) | Reported Result (WHAT) | Output Database (WHERE) | Category | Citation (Reference/Recital) |
| ETS Directive | EU-ETS | 13/10/03 | Directive 2003/87/EC (http://data.europa.eu/eli/dir/2003/87/oj (accessed on 23 January 2024)) | Annual Emission Report | – | Annex X Commission Implementing Regulation (EU) 2018/2066 (http://data.europa.eu/eli/reg_impl/2018/2066/oj (accessed on 18 December 2023)) | Mandatory attempts | Companies | Annually per installation | National competent authority | Total annual emissions/installation | – | Reorientation of capital flows by economic means | *"to contribute to fulfilling the commitments of the European Community (…) through an efficient European market in greenhouse gas emission allowances"* (Recital (5)) |
| The European Pollutant Release and Transfer Register | E-PRTR | 18/01/06 | Regulation (EC) No 166/2006 (http://data.europa.eu/eli/reg/2006/166/oj (accessed on 23 January 2024)) | – | – | Electronic Format according to Annex III Regulation (EC) No 166/2006 | Mandatory attempts | Companies | Annually per installation | European Energy Agency | Total annual emissions/installation | [49] | Facilitation of public participation | *"to facilitate public participation in environmental decision-making, as well as contributing to the prevention and reduction of pollution of the environment"* (Subject Matter) |
| European Environmental–Economic Accounts | EEEA | 06/07/11 | Regulation (EU) No 691/2011 (http://data.europa.eu/eli/reg/2011/691/oj (accessed on 23 January 2024)) | – | – | Electronic Format according to Annex I Regulation (EU) No 691/2011 | Mandatory attempts | Member State | Annually, per Member State | European Union/Eurostat | Annual emissions are divided into sectors and categories | [51] | Information provision for evidence-based policymaking | *"to provide high-quality statistics and accounts in the domain of the environment"* (Recital (4)) |
| Non-Financial Reporting Directive | NFRD | 22/10/14 | Directive 2014/95/EU (http://data.europa.eu/eli/dir/2014/95/oj (accessed on 23 January 2024)) | Global Reporting Initiative * | GRI | Sustainability Reports using Standards as stated in Article 5 of Communication from the Commission 2017/C 215/01 and similar standards | Mandatory attempts | Companies | Annually | Publication on the company website | Annual emissions/company | – | Individual | *"to raise to a similarly high level across all Member States the transparency of the social and environmental information provided by undertakings in all sectors"* (Recital (1)) |
| | | | | United Nations Global Compact * | UN Global Compact | | | | | | | | | |
| | | | | Organization for Economic Co-operation and Development (OECD) Guidelines for Multinational Enterprises * | OECD Guidelines | | | | | | | | | |

**Table 1.** *Cont.*

| Public Regulation—Descriptive Characterization | | | | Reporting Scheme—Descriptive Characterization | | | Historical Development | Methodological Characterization | | | | | Characterization of Desired Outcome (WHY) | |
|---|---|---|---|---|---|---|---|---|---|---|---|---|---|---|
| Name | Short Name | First Adopted | Legislative Document (Initial Version) | Name | Short Name | Technical Document (Currently Valid Version) | Phase | Reporting Entity (WHO) | Frequency (HOW OFTEN) | Recipient (TO WHOM) | Reported Result (WHAT) | Output Database (WHERE) | Category | Citation (Reference/Recital) |
| Effort Sharing Regulation | ESR | 30/05/18 | Regulation (EU) 2018/842 (http://data.europa.eu/eli/reg/2018/842/oj (accessed on 23 January 2024)) | National Energy and Climate Plan | NECP | Annex I of Commission Implementing Regulation (EU) 2022/2299 (http://data.europa.eu/eli/reg_impl/2022/2299/oj (accessed on 23 January 2024)) | Mandatory attempts | Member State | Every ten years, where necessary, update after five years | European Commission | Biennial emission reductions in non-ETS sectors | [52] | Monitoring for compliance | *"to fulfilling the Union's target"* and *"to achieving the objectives of the Paris Agreement"* (Subject Matter) |
| EU Taxonomy | – | 18/06/20 | Regulation (EU) 2020/852 (http://data.europa.eu/eli/reg/2020/852/oj (accessed on 23 January 2024)) | – | – | Commission Delegated Regulation (EU) 2021/2178 (Disclosures Delegated Act) Annex II (http://data.europa.eu/eli/reg_del/2021/2178/oj (accessed on 23 January 2024)) | New regulatory momentum | Companies | Per Product | Publication in line with the management report of Directive 2013/34/EU | Life-cycle emissions/activity along defined life-cycle phases | – | Reorientation of capital flows by economic means | *"to reorient capital flows towards sustainable investment"* (Recital (6)) |
| European Climate Law | – | 30/06/21 | Regulation (EU) 2021/1119 (http://data.europa.eu/eli/reg/2021/1119/oj (accessed on 4 February 2024)) | National Energy and Climate Plan | NECP | Annex I of Commission Implementing Regulation (EU) 2022/2299 (http://data.europa.eu/eli/reg_impl/2022/2299/oj (accessed on 23 January 2024)) | New regulatory momentum | Member State | Every ten years, where necessary, update after five years | European Commission | Strategies and measures designed to meet the objectives and targets of (…) the Union's climate-neutrality objective | [52] | Monitoring for compliance | *"ensure that both the Union and the Member States contribute to the global response to climate change as referred to in the Paris Agreement"* (Recital (8)) |
| Corporate Sustainability Reporting Directive | CSRD | 14/12/22 | Directive (EU) 2022/2464 (http://data.europa.eu/eli/dir/2022/2464/oj (accessed on 17 December 2023)) | European Sustainability Reporting Standard | ESRS | Annex I Commission Delegated Regulation (EU) 2023/2772 (http://data.europa.eu/eli/reg_del/2023/2772/oj (accessed on 23 January 2024)) | New regulatory momentum | Companies | Annually | Publication on the company website | Annual emissions/company | – | Individual | *"to set up a comprehensive Union framework on non-financial reporting that contains mandatory Union non-financial reporting standards"* (Recital (5)) |

**Table 1.** *Cont.*

| Public Regulation—Descriptive Characterization | | | | Reporting Scheme—Descriptive Characterization | | | Historical Development | Methodological Characterization | | | | | Characterization of Desired Outcome (WHY) | |
|---|---|---|---|---|---|---|---|---|---|---|---|---|---|---|
| Name | Short Name | First Adopted | Legislative Document (Initial Version) | Name | Short Name | Technical Document (Currently Valid Version) | Phase | Reporting Entity (WHO) | Frequency (HOW OFTEN) | Recipient (TO WHOM) | Reported Result (WHAT) | Output Database (WHERE) | Category | Citation (Reference/Recital) |
| Carbon Border Adjustment Mechanism | CBAM | 10/05/23 | Regulation (EU) 2023/956 (http://data.europa.eu/eli/reg/2023/956/oj (accessed on 23 January 2024)) | – | – | Annex I of Commission Implementing Regulation (EU) 2023/1773 (http://data.europa.eu/eli/reg_impl/2023/1773/oj (accessed on 23 January 2024)) | New regulatory momentum | Companies | Annually per installation | National competent authority | Total annual emissions/installation | – | Reorientation of capital flows by economic means | *"to prevent the risk of carbon leakage (…) and supporting the goals of the Paris Agreement"* (Subject Matter) |
| Ecodesign for Sustainable Products Regulation ** | ESPR | *** | COM/2022/142 final | Digital Product Passport | DPP | Annex III of COM/2022/142 final | New regulatory momentum | Companies | Per Product | *** | *** | *** | Individual | *"ensure a level playing field for products sold on the internal market"* and *"fostering sustainable product choices"* (Explanatory Memorandum) |
| Green Claims Directive ** | *** | *** | COM/2023/166 final | *** | *** | *** | New regulatory momentum | Companies | Per Claim | *** | *** | *** | Individual | enable *"consumers to take informed purchasing decisions".* (Recital (5)) |

* Examples only; further voluntary reporting standards may apply. ** Public regulation at the time of submission has not yet been legally adopted. Characterization is performed based on official EU proposals for the respective regulations and may be subject to change. *** Not available since public regulation at submission was not yet legally adopted.

3.1.3. Summary of Analytical Findings

From the historical review and the evaluation of key characteristics, we draw three analytical findings.

First, by building on well-established reporting concepts, formally self-regulatory reporting schemes (e.g., GRI) have triggered the development of mandatory reporting schemes and standards (such as the European Sustainability Reporting Standards). While this demonstrates that organizations behind these self-regulatory schemes have been able to exert a certain degree of influence on legislation, specific self-regulatory standards might lose relevance. Given the high level of activity within the EU, it is likely that this trend towards mandatory reporting schemes will continue. New self-regulatory schemes may emerge in areas where legislation has not yet defined specific reporting requirements. For instance, the private sector could play an active role in developing new best practices for obtaining supplier information and for reporting emissions along the value chain.

Second, both states and companies report GHG emissions for varying reasons. States primarily report to monitor compliance with intergovernmental agreements or to inform and involve the public in policymaking. While company-related reporting also aims to provide general information to the public, the desired outcomes of recent reporting schemes become increasingly more specific. The EU Taxonomy aims to "*reorient capital flows towards sustainable investment*" (Recital 6), and CSRD aims to "*set up a comprehensive Union framework on non-financial reporting*" (Recital 5). According to the ESPR and Green Claims proposal, they aim to "*enable sustainable consumer decisions*" (Explanatory Memorandum) and "*empower consumers to make informed purchasing decisions*" (Recital 5), respectively. These objectives imply that reporting requirements have gained political salience, which is underlined by the fact that they not only include an increasing number of reporting details but also become mandatory for an increasing number of companies.

Lastly, a development that is hard to predict is the indirect effect European regulations will have. On the one hand, this has an internal perspective as the EU Taxonomy pressures investors to steer away from fossil fuels. On the other hand, it is currently open to how far CBAM will lead to a global diffusion of regulatory standards or countermeasures from major economies.

*3.2. Analysis of Accounting Methodologies*

3.2.1. Classification and Review of Historical Development

Although GHG accounting encompasses a broad scope of single approaches with a confusing variety of designations, this diversity can be described by a limited number of fundamental methodologies classified along system boundaries (or scales). Accordingly, Stechemesser and Guenther [9] discern national, organizational, product, and project scale. Given the great methodological advancements made in GHG accounting within the last decade and many new reporting schemes and obligations introduced requiring correct and consistent GHG accounting, we critically review if their classification approach remains valid or requires refinements.

On the national scale, GHG accounting has evolved as an integral part of intergovernmental efforts to monitor and reduce territorial GHG emissions in line with the Kyoto Protocol. First published in 1994, the respective accounting methodology is today known as the "2019 Refinement to the 2006 IPCC Guidelines for National Greenhouse Gas Inventories" (IPCC-Guidelines) [53]. Methodologically, it is based on the key concept of territorial accounting, which implies that only activities within the national territory and offshore areas over which a country has jurisdiction are included. However, in line with the intergovernmental development of the SEEA framework, another methodological approach is defined in the "Manual for Air Emissions Accounts" (AEA) 2015 edition [54]. Unlike territorial accounting, residential accounting locates GHG emissions in the country where the emitting company's center of economic interest is located, regardless of the emission's geographical origin. This approach acknowledges the strong correlation between Gross

Domestic Product and GHG emissions, forming the foundation for an environmentally extended input–output (EEIO) analysis.

The organizational scale focuses on the legal definition of organizations and has evolved within the proactive and self-regulatory GHG accounting efforts of front-running organizations in the 1990s. First published in 2001, the GHG Protocol Corporate Accounting and Reporting Standard (Corporate Standard) divides organizational emissions into Scopes 1, 2, and 3 [55]. Guidance on how to account for Scope 3 emissions followed in 2011 within the "GHG Protocol Corporate Value Chain Accounting and Reporting Standard" (Scope 3 Standard) [56]. Even though the European Commission and other organizations have introduced further organizational accounting standards, such as the Organizational Environmental Footprint [57] or organizational LCA [58], the concept of the three scopes became so influential that the GHG Protocol Standards built the methodological basis for voluntary (e.g., GRI, UN Global Compact and OECD Guidelines) and mandatory reporting schemes (European Sustainability Reporting Standards). From the methodological perspective, a distinction should be acknowledged. Different from Scope 1 and 2, Scope 3 includes activities outside the organization's legally defined boundaries (e.g., the extraction of raw materials) and, thus, does not correspond to the system boundaries of the accounting organization. While this may also apply to Scope 2 emissions, companies have direct contractual relationships that enable them to exert direct influence. Therefore, we distinguish organizational Scope 1 and 2 from organizational Scope 3 accounting.

As the reporting-scheme analysis indicates, a scale below or within the organization should be discerned—the installation. Installation-based GHG accounting has its conceptual origins in the 1970s and depicts the fundamental methodological basis for the EU-ETS. After its legal adoption, the accounting methodology has been subject to constant revision and is currently defined within a Commission Implementing Regulation (EU) 2018/2066 (in the latter referred to as ETS-Guidelines) [59]. Installation-based inventories include direct emissions from the installation and indirect emissions caused by the supply of energy, monitored either based on a calculation or a measurement approach. While the calculation approach refers to emission factors or other specific conversion factors, the measurement approach requires continuous emission monitoring using sensors, meters, or other suitable equipment.

Product-based GHG accounting evolved from the scientific context of the life-cycle assessment (LCA), which originated in the 1970s and was defined in the ISO14040/14044 standards [60]. Today, LCA is used by research, governmental organizations, and companies to assess the environmental impacts of products and services throughout the entire life cycle, regardless of national boundaries. Due to constant methodological advancements, multiple variations exist, such as the Product Environmental Footprint (PEF) method [57] or specific LCA-based standards for GHG accounting [61,62]. Until recently, LCAs have mainly been conducted for internal decision-making, external voluntary disclosures (e.g., within EPDs), or scientific purposes; however, within the EU's new regulatory momentum, LCAs are increasingly required by legislation (e.g., within the EU Taxonomy).

The project scale has not yet been implemented into international GHG reporting schemes or legislation. And since projects account for a change in state in time, induced by a certain activity or measure that may occur at any other scale, we reject this scale.

In conclusion, our analysis discerns six fundamental methodologies: *territorial*, *residential*, *installation-based*, *organizational Scope 1 and 2*, *organizational Scope 3*, *and product-based accounting*.

### 3.2.2. Characterization

The characterization of the six accounting methodologies (Table 2) is derived along descriptive characteristics (e.g., name), methodological characteristics (e.g., system boundary), and the historical development phases derived in Section 3.1.1. Additionally, we include two further characteristics that significantly determine the workload of companies and the overall working process. First, we characterize the role of companies for the respective accounting methodology, discerning to what extent companies contribute to the inventory

calculation (workload for companies) and whether companies request or provide data to the accounting entity. Second, we characterize the data sources utilized within accounting. Here, we specify if data sources are entirely within the legal access of the accounting entity if data sources are conclusively specified by the accounting methodology, and which data sources are possibly consulted in practice.

3.2.3. Summary of Analytical Findings

The historical analysis reveals that accounting methodologies partially evolved as an integral part of reporting schemes (territorial, residential, and installation-based accounting) and partially within self-regulatory or scientific efforts (organizational and product-based accounting). This finding, on the one hand, indicates that accounting methodologies follow the historic development phases of reporting schemes and, on the other hand, explains why methodological variations only exist for organizational Scopes 1, 2, and 3 and product-based accounting. In contrast to the other accounting methodologies, legislators did not design these for distinct reporting needs; rather, they evolved within long and ongoing multi-stakeholder discussions, adapting the basic methodological concept to specific accounting peculiarities.

The data-related analysis indicates that companies contribute to territorial and residential accounting only on special requests; thus, their accounting workload remains minimal. For all other accounting methodologies, companies and other organizations are the main reporting entities. For these, several characteristics show a clear distinction between installation-based and organizational scope 1 and 2 accounting on one hand, and organizational scope 3 and product-based accounting on the other hand. The first group builds (almost) entirely on predominantly available internal data sources that can be feasibly measured or calculated. Electricity-related emissions are the only exception, which, however, must be disclosed by energy providers (Directive 2019/944). The second group is conceptually built on the life-cycle perspective, reaching beyond organizational or national boundaries. Thus, they require the extensive use of heterogenic data sources, which are not within the accounting entity's legal access but must be requested from suppliers or customers, derived from sustainability databases, obtained from the literature, estimated, assumed, or acquired similarly. Due to this overarching methodological feature, we propose to jointly denote organizational Scope 3 and product-based accounting as "*life cycle-based accounting methodologies*", albeit while acknowledging general differences in methodological procedures and required data. While Scope 3 accounting maintains a top–down perspective, requiring predominantly aggregated data for all companies' activities, LCA focuses on individual products, relating all results to one functional unit.

**Table 2.** Characterization of Accounting Methodologies.

| Accounting Level | Name | Short Name | Phase | Integral Part of Reporting Scheme | System Boundary | Methodological Document | Variations in Methodology | Accounting Entity | Reference Period | GHG Considered | Accounting Workload for Companies | Do Companies Request Data from Accounting Entity | Do Companies Provide Data to Accounting Entity | Data Sources Entirely within Legal Access | Data Sources Conclusively Specified | Internal Data | Supplier/Consumer Request | Open access/Statistics/Literature | Generic sustainability Databases/Model-Based | Estimations/Assumptions |
|---|---|---|---|---|---|---|---|---|---|---|---|---|---|---|---|---|---|---|---|---|
| Territorial Accounting | 2019 Refinement to the 2006 IPCC Guidelines for National Greenhouse Gas Inventories | IPCC-Guidelines | Intergovernmental | Yes | National borders (territory principle) | [53] | None | The competent authority of the Member State | 1 year | Defined in Volume 1, Chapter 1 | Deliver data on special request | No | Yes | No | Yes | | | X | | X |
| Residential Accounting | Air Emissions Accounts | AEA | Intergovernmental | Yes | National borders (Residence principle) | [54] | None | Member State | 1 year | $CO_2$, $CH_4$, $N_2O$, HFC, PFC, $SF_6$ | None | No | No | No | Yes | | | X | | X |
| Installation-based Accounting | Commission Implementing Regulation (EU) 2018/2066 | ETS-Guidelines | Mandatory attempts | Yes | Installation | [59] | None | Company | 1 year | Mainly $CO_2$, defined in Annex I Directive 2003/87/EC | Account full inventory | Yes | No | Yes | Yes | X | | | | X |
| Organizational Accounting | GHG Protocol Corporate Accounting and Reporting Standard | GHG Protocol Corporate Standard | Self-regulatory | No | Organization: Scope 1 and 2 | [55] | [57,58] * | Company | 1 year | $CO_2$, $CH_4$, $N_2O$, HFC, PFC, $SF_6$ | Account full inventory | Yes | No | Yes | Yes | X | | | | X |
| Organizational Accounting | GHG Protocol Corporate Value Chain (Scope 3) Accounting and Reporting Standard | Scope 3 Standard | Self-regulatory | No | Organization: Scope 3 | [56] | [57,58] * | Company | 1 year | $CO_2$, $CH_4$, $N_2O$, HFC, PFC, $SF_6$ | Account full inventory | Yes | Yes | No | No | | X | X | X | X |
| Product-based Accounting | Life-Cycle Assessment | LCA | Self-regulatory | No | Defined parts or entire life cycle | [60] | [57,61,62] * | Company, Science | Individually defined | All, if data is available | Account full inventory | Yes | Yes | No | No | X | X | X | X | X |

* Examples only; further standards or guidelines may apply.

## 4. Discussion

Our analysis allows us to assign public regulations and their corresponding reporting schemes to their underlying accounting methodology, enhancing Figure 1 to Figure 2.

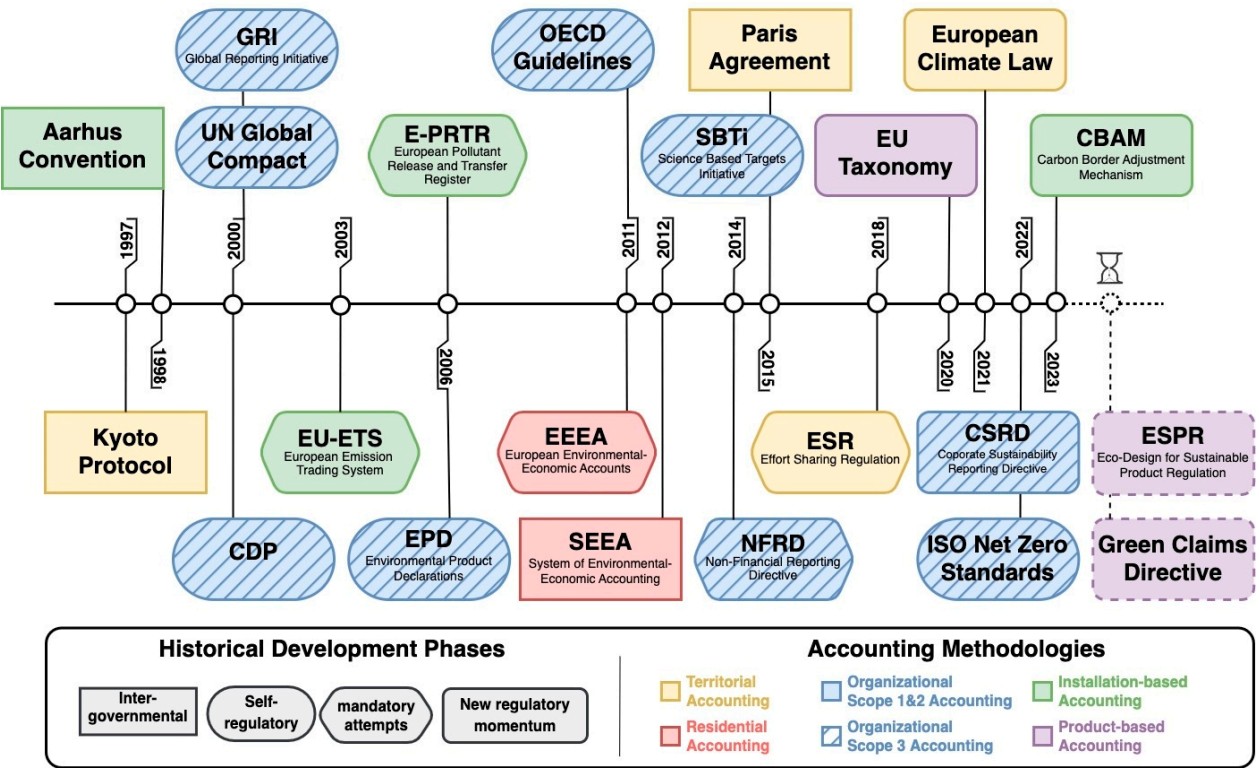

**Figure 2.** Assignment of reporting schemes and public regulations to their underlying accounting methodologies.

Based on this assignment, we discuss the interrelations between reporting schemes and accounting methodologies along three dimensions: the historical, the stakeholder, and the methodological dimension.

Within the historical dimension, there is a continuous interaction of reporting schemes and accounting methodologies that, however, over the course of time changed their character. Within the four identified historic development phases, first, intergovernmental agreements and their reporting schemes went along with customized accounting methodologies directly providing methodological guidance for respective reporting needs (territorial, residential, and installation-based accounting). Later, organizational and product-based accounting developed within self-regulatory and scientific activities, following the impetus from politics but not primarily serving political-implementation purposes. Since, in the subsequent phase, the EU implemented more mandatory attempts, these, in turn, influenced the voluntary reporting schemes that, however, still stayed independent of legislative reporting and its related accounting methodology. In contrast, within the EU's new regulatory momentum, current and upcoming public regulations broadly draw from prior methodological accounting advancements, shifting these from voluntary to mandatory application. In conclusion, the development of reporting and accounting exhibits a bi-directional relationship (Figure 3). Either public regulations established reporting schemes, which specified customized accounting methodologies, or, vice versa, existing accounting methodologies triggered the development of new reporting schemes and regulations. Since public regulations have historically been the predominant driver for reporting advancements, they will foreseeably continue to shape and drive future methodological accounting developments.

**Figure 3.** Linear, bi-directional relationship of public regulations, reporting schemes, and accounting methodologies.

Within the stakeholder dimension, a state-to-company and a company-to-company relationship can be discerned. On the one hand, the more governments risk missing their national and intergovernmental reduction targets, the more they increase the reporting pressure on companies. In the EU, this is evident not only by the rising number of reporting obligations but also by comparing the desired outcomes of earlier and more recent regulations. While earlier outcomes were formulated generally (e.g., to inform the public), recently desired outcomes have become significantly more specific (e.g., to reorient capital flows). Thus, today's regulations aim to actively direct investor and customer decisions toward reducing GHG emissions, ultimately affecting companies' financial and market-related interests. Overall, the EU induces increasing indirect but effective pressure on companies by demanding specific information disclosure. On the other hand, the shift towards specific outcomes is accompanied by the increasing use of life-cycle-based accounting methodologies, which demand two-directional information exchange across the value chain. While companies must obtain supplier information for their own reporting, they must also provide information to other firms. Therefore, the regulatory pressure is complemented by increasing pressure along the supply chain, even for small and medium enterprises that are not (yet) subject to mandatory reporting.

The methodological dimension builds upon two significant observations from the historical and stakeholder dimensions. First, the historical dimension shows that mandatory reporting schemes have picked up existing voluntary accounting methodologies; however, they have not been adapted to the requirements for mandatory use, especially regarding standardized methodological procedures and data sources for transparent and reproducible assessments. Thus, a finding from an early publication remains valid: "*Methods and standards used to collect and report (...) data are unclear and lack uniformity. Without a uniform and regularized system of measurement and third-party verification of data, valuable comparisons are difficult to make*" [23] (p. 336). Consequently, companies are required to make their own methodological or data-related decisions which compromises the comparability, reproducibility, and quality of the results, ultimately also affecting its legal certainty. Second, the increased demand for external data from the value chain further intensifies the accounting workload and comes with a major regulatory drawback. Since individual companies cannot be held legally responsible for the quality of external data, life-cycled approaches may be highly suitable for information or monitoring purposes but not for direct regulation of single companies. This direct regulation is not yet part of public regulations; however, the prominent position of life-cycle approaches in European policies suggests future developments in this direction (e.g., within ESPR), calling for introducing a legal framework for standardizing data sources.

Above that, a general methodological conflict arises due to the co-existence of two specific political aims: the reduction in national and EU-wide GHG emissions, respectively, to comply with intergovernmental agreements, and the prevention of burden shifting (such as carbon leakage) to non-EU countries. While the life-cycle-based accounting methodologies conceptually build upon upstream and downstream data, independent of their geographic location or legal ownership, the attempt to add up results nationally or within the EU is methodologically unfeasible due to the double counting of emissions. Although this issue was acknowledged in the literature long before [63–65], it was of minor practical relevance since national and voluntary life-cycle-based accounting co-existed mainly independently. However, today's prominent use of life-cycle-based methodologies shifts this conflict into

the center of the political arena, connected with the issue to clarify which consequences may result from the attribution of reduction efforts to stakeholders in the value chain [66–68].

Overall, the current phase of the EU's new momentum depicts a quantitative growth in reporting schemes and a qualitative change by broadly incorporating accounting methodologies that were originally not intended for mandatory purposes. From the company perspective, as long as methodological and data-related choices remain undefined, firms will foreseeably stay reluctant to disclose data for fear of performing worse than competitors due to diverging methodological assumptions or data sources. Acknowledging the importance of harmonization, the EU introduced the first harmonization attempts, e.g., by replacing NFRD with CSRD, along with mandatory sustainability reporting standards. However, they lack suggestions on harmonizing Scope 3 data acquisition. To this end, high hopes lie in the digital product passport, which will be introduced as part of the upcoming ESPR, whose methodological peculiarities are yet to be defined.

## 5. Conclusions

Overall, GHG reporting must be evaluated to determine whether it goes beyond simply meeting reporting obligations or achieves real results in terms of climate mitigation. Here, the quality of carbon information is a crucial prerequisite. Thus, our findings can support efforts from policymakers, industry, and academia to ensure quality and transparency regarding this information. The analysis of GHG reporting and GHG accounting interactions under the EU policy framework provides new insights into methodological and data challenges. These insights are valid not only for the EU but also for a generic understanding of interaction mechanisms. Specifically, in terms of the current multi-objective intentions of EU policies, to foster pressure on companies to fulfill national (and, respectively, EU) GHG mitigation targets and to prevent burden-shifting, additional actions will be needed to tailor accounting methodologies in line with the desired outcomes of reporting obligations. As an overarching issue, the intent to manage data and information exchange in companies' (global) value chains demands addressing data ownership issues and allocating total GHG budgets to individual stakeholders.

Independent of the specific details of future reporting obligations, the ongoing policy momentum will drive the demand for companies to disclose more specific carbon information on individual processes and products. Thus, companies will necessarily have to develop additional or novel approaches deriving data from existing internal sources or sound theoretical modeling approaches. By contributing to sophisticating the overall theoretical understanding of current GHG reporting and accounting practices, the findings of our analysis may support the development of systematic and transparent data acquisition and management procedures. Sound and theoretically well-founded data-management concepts may be the basis for taking benefit from the inclusion of new information technologies, such as artificial intelligence. These are highly promising to foster efficiency and reduce the workload of data management [69], but they have not yet been systematically explored under the theoretical framework of European GHG reporting.

**Author Contributions:** Conceptualization: J.B. and L.S.; methodology: J.B. and L.S.; validation: F.Z., M.L. and G.S.; investigation: J.B.; writing—original draft preparation: J.B. and F.Z.; writing—review and editing: M.L., G.S., and L.S.; visualization: J.B. and M.L.; supervision: M.L. and L.S.; funding acquisition: L.S. and M.L. All authors have read and agreed to the published version of the manuscript.

**Funding:** The research of TU Darmstadt was conducted within the Merck Sustainability Hub 2021–2023, a joint research platform of Merck and TU Darmstadt.

**Institutional Review Board Statement:** Not applicable.

**Informed Consent Statement:** Not applicable.

**Data Availability Statement:** No new data were created or analyzed in this study. Data sharing does not apply to this study.

**Acknowledgments:** The authors would like to thank all members of the Merck Sustainability Hub for the ongoing interactive and lively discussions, which were a great help in deriving the concepts for this study.

**Conflicts of Interest:** The authors declare no conflicts of interest. The funders had no role in the design of the study; in the collection, analyses, or interpretation of data; in the writing of the manuscript; or in the decision to publish the results.

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
