# Peer review of "Back in the Driver’s Seat: How New EU Greenhouse-Gas Reporting Schemes Challenge Corporate Accounting"

_sustainability, doi:10.3390/su16093693_

Round 1

Reviewer 1 Report

Comments and Suggestions for Authors

The inclusion of the word 'challenge' is not clearly justified in this ms. The collation of facts concerning the evolution of reporting schemes may be of interest but most of the information in Tables is not directly relevant to the text. Tables 1 and 2 should be greatly shortened to include points included in the text alone. There is too much repetition re definitions of reporting and accounting in pages 3 and 4 and again in Discussion. The Summary is not informative, there is little analysis and evaluation. Maybe better to refer back to the aims of the study. Conclusions are brief and repetitive and the intended meaning of the last sentence in unclear to me. AI is introduced for the first time, with assertions without references. References are incomplete: give addresses for websites and dates accessed. Standardise citing of journal titles: either all with full title or all with abbreviated title. Standardise use of capital letters. You need to have the ms edited by an English speaker as there are many words which are not appropriate within sentences, and sentences for which the intended meaning is not clear

Points to consider:

What about city and community scale accounting? eg Covenant of Mayors,  ICLEI. Might these provide guidance for companies? You include little on the evolution of methods for accounting, yet you argue that accounting and reporting are equally important. 

How are we to identify fraud in data provided by companies? Verification methods are crucial

Scope 3 emissions are very complex, and data for many are unlikely to be available, and it will be hard to identify a feasible standardised method for accounting. Your assumption in 3.1.3 seems rather optimistic, and is not referenced. 

What is your view on the likelihood of developing an EU or global scheme for accounting? Which takes into account that Scope 3 emissions may be greater than those for Scopes 1 and 2.

Comments on the Quality of English Language

Extensive editing by an English speaker needed as there are inappropriate words in many sentences, and inclusion of 'words' which are not in fact words used in technical English, leading in some instances to sentences and paragraphs which do not have clear meaning

Reviewer 2 Report

Comments and Suggestions for Authors

Minor revision

This paper focus on the application of an analytical approach for GHG accounting and reporting in response to stakeholder/policy expectations. The authors classify key representatives of the tasks related to the calculation and reporting and analyze their specific characteristics. Finally, the authors discuss about the assignment of public regulations to accounting methodologies based on three aspects.

The paper certainly meets the aim and the scope, as well as, the high academic standards of the Sustainability Journal. However, the following specific improvements should be made, before accepting the paper for possible publication to the Journal.

Please do not use first person in the text. Please revise the manuscript accordingly.

In the introduction section, in lines 66-82, I claim that the authors could provide some comparisons between the published works in the years 1994 and 2022.

In the materials and methods section, could you please specify the reason why the authors do not include others indicators (social, governance) on their study? It would be also interesting to include a short paragraph in which the authors could provide in brief the potential influence of the others indicators in the results (reporting).

Comments on the Quality of English Language

Minor editing of English language required. Please do not use first person. 

Reviewer 3 Report

Comments and Suggestions for Authors

This work focus on the GHG reporting and GHG accounting history development, characterization and intercorrelation under the EU regulations. Overall, the work is well designed and written. But there are some issues for the authors to address:

1) introduction is too long, If possible, please shorten the introduction to one and a half page at most.

2) line 75, Guenther [9] state that carbon accounting definitions. Grammar error, better to use can be defined as; Line 141, First, according to [12], better if use accordingly [12],

3) For figure 1, caption is not proper. Most of the regulations in Figure 1 are based on EU laws and rules only, please specify this as some of them are not applicable in US or Asia. That may mislead readers.

4) Figure 1 and figure 2 almost the same except the different color highlighted accounting methodologies, why combine them together like Figure 2?

Comments on the Quality of English Language

The work is well written with only minor corrections needed:

1) introduction is too long, If possible, please shorten the introduction to one and a half page at most.

2) line 75, Guenther [9] state that carbon accounting definitions. Grammar error, better to use can be defined as; Line 141, First, according to [12], better if use accordingly [12]

Round 2

Reviewer 1 Report

Comments and Suggestions for Authors

Greatly improved ms. I still think you should mention the key role of verification but perhaps there is no room for it in this ms. Maybe in your next paper?